# Flow and Heat Transfer of CoFe_2_O_4_-Blood Due to a Rotating Stretchable Cylinder under the Influence of a Magnetic Field

**DOI:** 10.3390/bioengineering11040317

**Published:** 2024-03-26

**Authors:** Jahangir Alam, Ghulam Murtaza, Efstratios E. Tzirtzilakis, Shuyu Sun, Mohammad Ferdows

**Affiliations:** 1Research Group of Fluid Flow Modeling and Simulation, Department of Applied Mathematics, University of Dhaka, Dhaka 1000, Bangladesh; jahangircu1994@gmail.com; 2Department of Mathematics, Comilla University, Cumilla 3506, Bangladesh; murtaza@cou.ac.bd; 3Fluid Mechanics and Turbomachinery Laboratory, Department of Mechanical Engineering, University of the Peloponnese, 22100 Tripoli, Greece; etzirtzilakis@uop.gr; 4Physical and Engineering Division, King Abdullah University of Science and Technology, Thuwai 23955-6900, Saudi Arabia; shuyu.sun@kaust.edu.sa

**Keywords:** biomagnetic fluid dynamics, rotational viscosity, swirling flow, magnetic field, blood, magnetic particles, stretching cylinder, finite difference method

## Abstract

The flow and heat transfer of a steady, viscous biomagnetic fluid containing magnetic particles caused by the swirling and stretching motion of a three-dimensional cylinder has been investigated numerically in this study. Because fluid and particle rotation are different, a magnetic field is applied in both radial and tangential directions to counteract the effects of rotational viscosity in the flow domain. Partial differential equations are used to represent the governing three-dimensional modeled equations. With the aid of customary similarity transformations, this system of partial differential equations is transformed into a set of ordinary differential equations. They are then numerically resolved utilizing a common finite differences technique that includes iterative processing and the manipulation of tridiagonal matrices. Graphs are used to depict the physical effects of imperative parameters on the swirling velocity, temperature distributions, skin friction coefficient, and the rate of heat transfer. For higher values of the ferromagnetic interaction parameter, it is discovered that the axial velocity increases, whereas temperature and tangential velocity drop. With rising levels of the ferromagnetic interaction parameter, the size of the axial skin friction coefficient and the rate of heat transfer are both accelerated. In some limited circumstances, a comparison with previously published work is also handled and found to be acceptably accurate.

## 1. Introduction

The study of physical problems involving rotating stretchable cylindrical geometries is possibly the most well-known and famous research subject in fluid mechanics due to their versatile applications in biomedical, engineering, and industry, including cancer treatment, drug delivery, magnetic resonance imaging (MRI), jet motors, turbine systems, and hard disks. A growing amount of fluid dynamics, particularly biofluid mechanics, is being used to assist with clinical pathology diagnosis and treatment decision making. Blood was used in this investigation as a biofluid. The investigation of the connection between pathophysiological observations and blood flow phenomena is made easier by ongoing developments in computational model capabilities, imaging modalities, and measuring methods. In conjunction with underlying physiological models, a potent suite of instruments is made possible to address unfulfilled clinical demands, namely in the areas of improved diagnosis, evaluation, and treatment outcome prediction. The link between clinical findings and blood biomechanics is the subject of the current paper: biofluid with magnetic particles. In addition to a basic interest in these connections, there is a substantial practical benefit: studying immediately accessible or computed biofluidic observables can help with the assessment and even prediction of treatment effects, in addition to better diagnostic skills. The mathematical topic that is being addressed also examines the consistency between cardiovascular pathophysiology and methodologies, as well as the measurement and computation of relevant biophysical parameters. We hope that our research will help people to better understand blood flow, particularly in relation to medical applications like MRI, CT, and ultrasound that are used for diagnostic and therapeutic purposes. By injecting magnetic particles into the bloodstream, this could also be helpful in the delivery of drugs to particular body parts while taking particular magnetic field ranges into consideration.

Von Karman’s [1] groundbreaking study on a rotating disk has attracted a lot of attention and has been thoroughly examined by numerous researchers. The first person to introduce fluid flow over a stretched plate was Crane [2]. Later, Ming et al. [3] conducted a numerical analysis of heat transport for an incompressible power law fluid flow over a stretching sheet. Wang [4] later obtained a solution to the momentum equation for a three-dimensional flow across a stretching disk. Fang [5] expanded Karman’s [1] work on the stretchable revolving disk and the stretchable stationary disk. Later, Fang and Tao [6] conducted an analysis of the stretching phenomenon caused by an unsteady flow of viscous fluid across a stretchy spinning disk. Magnetohydrodynamics (MHD) flow with the impact of suction or injection over a nonlinear stretching surface was presented by Kudenatti et al. [7]. Hayat et al. [8] explored three-dimensional MHD flow with viscous dissipation and joule heating. Using the bvp4c approach, which is a finite difference code that implements the three-stage Lobatto IIIa formula in MATLAB (Version 2018b), Khan et al. [9] studied the steady SiO_2_-MoS_2_/water hybrid nanofluid flow induced by a rotating and stretching cylinder. They found that azimuthal and wall stress characteristics decrease with increasing values of Reynolds number, whereas heat and flow distributions increase as the volume percentage of solid particles increases. Jusoh et al. [10] studied the duality behavior of an MHD swirling ferrofluid induced by a stretched sheet using the bvp4c function approach in MATLAB. Zhang et al. [11] used the Cattaneo–Christov theory to explore the transfer properties of SiO_2_-MoS_2_/water hybrid viscous nanofluid flow through a rotating and stretching cylinder under the influence of the Lorentz force and heat source/sink. Ahmad et al. [12] demonstrated the boundary layer movement of viscous fluid caused by a spinning stretched disk under the influence of a magnetic field. The effects of Hall current and slip parameters were investigated by Krishna et al. [13] for the physical problem of the MHD rotating elastico-viscous fluid in a porous medium. 

Research on improving heat transmission utilizing magnetic and non-magnetic particles has recently attracted a lot of interest in fluid mechanics, particularly in the field of biofluids. The study of the effects of magnetic fields on fluid dynamics is relatively recent in the subject of biomagnetic fluid dynamics. Nanofluid is a general term for the suspension of nanoparticles in basic fluids like water, blood, oil, etc. Due to the greater thermal conductivity of nanoparticles when compared to base fluid, as mentioned by Choi et al. [14], this topic is of significant interest. Recently, with the use of the Runge–Kutta–Fehlberg numerical technique, Wang et al. [15] examined the impact of the slip condition on three-dimensional boundary layer flow across a stretched surface. Using a computational simulation of MHD stagnation point flow over a stretching sheet, Ravindra et al. [16] examined how to best balance the effects of thermal radiation, joule heating, and heat source/sink. The constant TiO_2_-Cu/H_2_O fluid flow over a heated stretched surface with varying thermal conductivity was investigated by Shah et al. [17]. Using the shooting technique, they discovered that when nanoparticles are suspended in engine oil, wall shear stress is reduced and the rate of heat transfer is accelerated. The behavior of non-Newtonian fluid flow across a stretching surface under the combined influence of magnetic and electric fields was quantitatively investigated by Bafakeeh et al. [18]. In their study of Fe_3_O_4_-CoFe_2_O_4_/water hybrid nanofluid flow in a Darcy porous medium with mixed convection flow, Zainodin et al. [19] took into account the impact of higher-order chemical reactions as well as slip conditions. Alam et al. [20] studied the combined effects of magnetohydrodynamics and ferrohydrodynamics (FHD) on biomagnetic fluid under the influence of a magnetic dipole for steady blood flow with Fe_3_O_4_ magnetic particles. They found that blood loss in biomagnetic fluid dynamics (BFD) cases is noticeably less than in MHD and/or FHD instances. Similar attempts were discovered in the investigation by Ferdows et al. [21]. The distinction between the two studies is that [21] examined an unsteady flow scenario using CoFe_2_O_4_ as the magnetic particles. By utilizing the concept of response surface methodology in order to scrutinize the physical parameters, Khashi’ie et al. [22] examined the unsteady flow of Fe_3_O_4_-CoFe_2_O_4_/water while considering the effects of heat generation through a rotating disk. Souayeh et al. [23] studied the impacts of activation energy, gyrotactic microorganisms, and electromagnetohydrodynamics on Au-Ag/blood flow in a symmetric peristaltic channel. Dinarvand et al. [24] discussed the incompressible viscous blood-based hybrid nanofluid flow using the Jeffery–Hamel problem through the converging/diverging channel under slip effects. Magnetic field effects on flow and heat transfer of water-based nanofluid over an extending cylinder have been examined by Ashorynejad et al. [25]. The authors considered Cu, Ag, Al_2_O_3,_ and TiO_2_ as nanoparticles. Using the Runge–Kutta technique in numerical simulations, they observed that a higher performance of cooling systems is obtained for Cu particles when low values of magnetic parameters are considered, and using Al_2_O_3_ particles gives the highest performance when the magnetic parameters are considered the highest. The nanofluid flow and heat characteristics in a rotating system between two horizontal plates were discussed by Sheikholeslami et al. [26]. They noted that the rate of heat transfer augmented with the larger values of the particle volume fraction and suction/injection parameter, while it declined with variation in the rotation parameter. Alghamdi et al. [27] studied the properties of nano- and hybrid nanofluid flow between two permeable channels, taking into account the MHD, heat source/sink, Cu/blood, and Cu-CuO/blood. Ali et al. [28] replicated the MHD peristaltic pumping of non-Newtonian blood flow with copper and gold nanoparticles. It was observed that there is a strong opposition to blood flow at the ciliated micro-vessel wall due to a thin electric double layer and a larger helping electric body force. Maqbool et al. [29] carried out a mathematical investigation of the Lorentz force and nanoparticles, such as copper, in the flow of tangent hyperbolic blood via a ciliated tube. They noticed that the presence of a magnetic field causes blood flow to quicken, which may also aid in improving heat transfer. Kumar et al. [30] provided a mathematical model of blood flow with single-wall and multi-wall carbon nanotubes across a spinning disk, together with magnetic field and heat source effects for medical applications such as drug administration and cancer treatment. They discovered that the blood multiwall carbon nanotube heat profile is substantially more dominant than the blood single-wall carbon nanotube heat profile. MHD hybrid thermal and flow performance fluid flow over a stretching sheet was examined by Qayyum et al. [31]. In that work, uranium dioxide nanoparticles were hybridized with copper, copper oxide, and aluminum oxide, and blood was treated as the base fluid. 

The abovementioned authors have discussed and presented their analysis of regular fluid, nanofluid, and hybrid nanofluid heat and flow characteristics in various geometries from the aforementioned studies and also noted that there are very few studies on the direction of nanofluid flows in rotating stretchable cylinders to investigate the biomagnetic fluid with magnetic particles (blood-CoFe_2_O_4_). This inspiration led to the current research, which focuses on the movement of blood containing magnetic particles as it passes through a stretchable rotating cylinder while being exposed to an applied magnetic field that is directed in the tangential and radial directions of the cylinder, respectively. Through the use of the common finite difference method, non-dimensional, highly nonlinear ODEs are generated. The results are then depicted in visual scenarios. The present outcomes have several potential applications in medical and engineering sectors, including drug administration, cancer treatment, and magnetic resonance imaging, etc. Keeping in mind the aforementioned applications of magnetic fluids and flow caused by a rotating stretchable cylinder surface, the main objectives and novelty of this study are:To investigate the flow and thermal characteristics of biomagnetic fluid that contains magnetic particles.The rotating stretching cylinder is used to check the flow analysis.The ferrohydrodynamics principle is taken into consideration when presenting the mathematical formulation.Strong magnetic field effects are discussed in light of their physical characteristics.To obtain the solutions for the present governing problem, a two-point boundary value technique based on a common finite difference scheme is used for the numerical treatment.

## 2. Mathematical Model with Flow Geometry

In this study, as depicted in Figure 1, we considered a steady, incompressible, and three-dimensional boundary layer axial flow towards a rotating stretchable cylinder in a blood-based magnetic ferrofluid (blood-CoFe_2_O_4_). The geometry of the physical model is shown in Figure 1, considering the following assumptions:$\left(w,v,u\right)$ are the fluid velocity components along the axial, tangential, and radial directions of the cylindrical surface, respectively, i.e., $\left(z,\phi ,r\right)$. The radius of a cylinder is R and its surface are stretched along an axial direction with velocity w=az. The cylinder rotates with a constant angular velocity ω about the axis r=0.Due to axisymmetric flow, the variation with respect to ϕ coordinate is ignored. The temperature of surface is constant, and it is about $T_{w}$, while the fluid temperature far away from the sheet is $T_{c}$, such that $T_{w}<T_{c}$. An external magnetic field is applied in the radial and tangential directions. As a result, the rotational velocity of magnetic particles in the base fluid is different from the vortices in the flow domain.

Considering the above assumptions and extending the work of [6,9,32,33,34,35], the conservation of mass, momentum, and energy equations are expressed as: (1)$$\frac{\partial u}{\partial r}+\frac{u}{r}+\frac{\partial w}{\partial z}=0$$
(2)$$u\frac{\partial w}{\partial r}+w\frac{\partial w}{\partial z}=\frac{\mu_{mf}}{\rho_{mf}}\left(1+\frac{3}{2}\varphi \;m\right)\;\left(\frac{\partial^{2}w}{\partial r^{2}}+\frac{1}{r}\frac{\partial w}{\partial r}\right)-\frac{\mu_{0}}{\rho_{mf}}M\frac{\partial H}{\partial z}$$
(3)$$u\frac{\partial v}{\partial r}+w\frac{\partial v}{\partial z}+\frac{uv}{r}=\frac{\mu_{mf}}{\rho_{mf}}\left(1+\frac{3}{2}\varphi \;m\right)\;\left(\frac{\partial^{2}v}{\partial r^{2}}+\frac{1}{r}\frac{\partial v}{\partial r}-\frac{v}{r^{2}}\right)-\frac{\mu_{0}}{\rho_{mf}}\frac{M}{r}\frac{\partial H}{\partial \phi }$$
(4)$$\begin{array}{ll} {\left(\rho C_{p}\right)}_{mf\;}\left(u\frac{\partial T}{\partial r}+w\frac{\partial T}{\partial z}\right)+\mu_{0}\;T\frac{\partial M}{\partial T}\left(w\frac{\partial H}{\partial z}+\frac{v}{z}\frac{\partial H}{\partial \phi }\right) & =\kappa_{mf}\left(\frac{\partial^{2}T}{\partial r^{2}}+\frac{1}{r}\frac{\partial T}{\partial r}\right)\\ & +\mu_{mf}\left[{\left(\frac{\partial u}{\partial r}-\frac{v}{r}\right)}^{2}+{\left(\frac{\partial w}{\partial r}\right)}^{2}\right] \end{array}$$

With corresponding boundary conditions: (5)$$\begin{array}{l} at\;\;\;r=R:\;\;\;u\;\left(r,z\right)=0,\;v\;\left(r,z\right)=\omega r\;,\;w\;\left(r,z\right)=0,T\left(r,z\right)=T_{w}\\ as\;\;\;r\to \infty :\;\;\;\;v\;\left(r,z\right)=0\;,\;w\;\left(r,z\right)=0,T\left(r,z\right)=T_{c} \end{array}$$
where $\rho ,\mu ,\mu_{0},\kappa$ are the biomagnetic fluid density, dynamical viscosity, magnetic permeability, and thermal conductivity, respectively. H is the applied magnetic field and the magnetization is M. Due to the application of the external magnetic field, in the flow domain, a rotation viscosity is observed that is identified by the term $\frac{3}{2}\mu_{mf}\;\varphi \;m$, where φ is the particle volume fraction and m is the effective magnetic number. T is the fluid temperature. The subscript symbol ${\left(\;\right)}_{mf}$ is noted for the meaning of magnetic fluid, i.e., suspension of magnetic particles with base fluid. 

The mathematical expression of the magnetic field is given by [36] as:(6)$$H=-\nabla \;\left(\frac{\gamma }{2\pi }\frac{\cos\;\phi }{r}\right)$$
where the scalar potential due to applied magnetic field is $\psi =\frac{\gamma }{2\pi }\frac{\cos\;\phi }{r}$.

After analogous manipulations of Equation (6), components of the magnetic field strength intensity in the radial and tangential directions can be written as: (7)$$\left\{\begin{array}{l} H_{r}=-\frac{\partial \psi }{\partial r}=\frac{\gamma }{2\pi }\frac{\cos\;\phi }{r^{2}}\\ H_{\phi }=-\frac{\partial \psi }{\partial \phi }=\frac{\gamma }{2\pi }\frac{\sin\;\phi }{r} \end{array}\right.$$

Therefore, the magnitude of the total magnetic field intensity is:(8)$$H=\sqrt{H_{r}^{2}+\frac{1}{r^{2}}H_{\phi }^{2}}=\frac{\gamma }{2\pi }\frac{1}{r^{2}}$$

By solving (8), the changes in magnetic field intensity in radial and tangential direction are given by:(9)$$\left\{\begin{array}{l} \frac{\partial H}{\partial r}=-\frac{\gamma }{\pi }\frac{1}{r^{3}}\\ \frac{\partial H}{\partial \phi }=0 \end{array}\right.$$

To describe the variation in the magnetization of the fluids in the flow domain, researchers have proposed several mathematical expressions involving temperature and/or magnetic field intensity. The following expression has been utilized in this study, according to [20,21]:(10)$$M=K\left(T_{c}-T\right)$$
where $T_{c}$ is the Curie temperature, and K is the well-known constant Pyromagnetic coefficient. Note that far away from the cylinder, the magnetization is zero since the ambient temperature is considered as $T_{c}$.

The theoretical model for the magnetic fluid properties is described in Table 1. Note that the subscripts symbol ${\left(\;\right)}_{f}$ identifies the regular fluid, and ${\left(\;\right)}_{s}$ identifies the magnetic particles. 

## 3. Similarity Transformations

The following similarity variables are introduced to Equations (1)–(5):(11)$$\eta \;=\frac{r^{2}-R^{2}}{2R}\sqrt{\frac{a}{\upsilon_{f}}},\;u=-\sqrt{a\upsilon_{f}}\frac{R}{r}f\left(\eta \right),\;v=\omega Rg\left(\eta \right),\;w=azf^{\prime }\left(\eta \right),\;\theta \left(\eta \right)=\frac{T_{c}-T}{T_{c}-T_{w}}$$

The continuity of Equation (1) is identically satisfied, and the set of Equations (1)–(4), using (11), is transformed to: (12)$$\begin{array}{l} \left(1+2\eta D\right)\;\left(1+\frac{3}{2}\varphi \;m\right)\;f^{\prime \prime \prime }+\left[2D\;\left(1+\frac{3}{2}\varphi \;m\right)+{\left(1-\varphi \right)}^{2.5}\;\left(1-\varphi +\varphi \;\frac{\rho_{s}}{\rho_{f}}\right)\;f\right]\;f^{\prime \prime }\\ \;\;\;\;\;\;\;\;\;\;\;\;\;\;\;\;\;\;\;\;\;\;\;\;\;\;\;\;\;\;\;\;\;\;\;-{\left(1-\varphi \right)}^{2.5}\;\left(1-\varphi +\varphi \;\frac{\rho_{s}}{\rho_{f}}\right)\;\;{f^{\prime }}^{2}+{\left(1-\varphi \right)}^{2.5}\beta \theta =0 \end{array}$$
(13)$$\begin{array}{l} {\left(1+2\eta D\right)}^{2}\;\left(1+\frac{3}{2}\varphi \;m\right)\;g^{\prime \prime }+\left[2D\;\left(1+\frac{3}{2}\varphi \;m\right)\;\left(1+2\eta D\right)+{\left(1-\varphi \right)}^{2.5}\;\left(1-\varphi +\varphi \;\frac{\rho_{s}}{\rho_{f}}\right)\;\left(1+2\eta D\right)\;f\right]\;g^{\prime }\\ \;\;\;\;\;\;\;\;\;\;\;\;\;\;\;\;\;\;\;\;\;\;\;\;\;\;\;\;\;\;\;\;\;+\left[{\left(1-\varphi \right)}^{2.5}\;\left(1-\varphi +\varphi \;\frac{\rho_{s}}{\rho_{f}}\right)\;Df-\left(1+\frac{3}{2}\varphi \;m\right)\;D^{2}\right]g=0 \end{array}$$
(14)$$\begin{array}{l} \left(1+2\eta D\right)\;\theta^{\prime \prime }+\left[2D+\frac{\kappa_{f}}{\kappa_{mf}}\left(1-\varphi +\varphi \frac{{\left(\rho C_{p}\right)}_{s}}{{\left(\rho C_{p}\right)}_{f}}\right)\mathrm{Pr}f\right]\theta^{\prime }-\frac{\kappa_{f}}{\kappa_{mf}}\beta \;Ec_{2}f^{\prime }\theta \\ +\frac{\kappa_{f}}{\kappa_{mf}}\left[\beta \;\varepsilon \;\lambda \;f^{\prime }-\frac{\left(1+2\eta D\right)}{{\left(1-\varphi \right)}^{2.5}}Ec_{1}{\left\{g^{\prime }-\frac{D}{\left(1+2\eta D\right)}g\right\}}^{2}-\frac{\left(1+2\eta D\right)}{{\left(1-\varphi \right)}^{2.5}}\;Ec_{2}{f^{\prime \prime }}^{2}\right]=0 \end{array}$$

The corresponding set of Equation (5) is also transformed to: (15)$$\begin{array}{l} f\left(0\right)=0\;,\;f^{\prime }\left(0\right)=1\;,\;g\left(0\right)\;=1\;,\;\theta \left(0\right)=1\\ g\left(\infty \right)\;=0\;,\;f^{\prime }\left(\infty \right)=0,\;\theta \left(\infty \right)=0 \end{array}$$

In (12)–(14), the following dimensionless parameters arise, known as: 

Ferromagnetic interaction parameter $\beta =\frac{\gamma }{\pi }\frac{\mu_{0}K\left(T_{c}-T_{w}\right)}{\rho_{f}a^{2}zr^{3}}$.

Temperature parameter $\varepsilon =\frac{T_{c}}{T_{c}-T_{w}}$.

Curvature parameter $D=\sqrt{\frac{\upsilon_{f}}{aR^{2}}}$.

Prandtl number $\mathrm{Pr}=\frac{{\left(\mu C_{p}\right)}_{f}}{\kappa_{f}}$.

Viscous dissipation parameter $\lambda =\frac{{\vartheta_{f}}^{3/2}\rho_{f}a^{3/2}z}{r\kappa_{f}\left(T_{c}-T_{w}\right)}$.

Eckert number due to rotation of the cylinder $Ec_{1}=\frac{\mu_{f}\omega^{2}R^{2}}{\kappa_{f}\left(T_{c}-T_{w}\right)}$.

Eckert number due to stretching of the cylinder $Ec_{2}=\frac{\mu_{f}a^{2}z^{2}}{\kappa_{f}\left(T_{c}-T_{w}\right)}$.

## 4. Physical Quantities of Interest

Two important physical terms of interest are the shear stress and the rate of heat transfer. 

**Shear stresses:** The axial and tangential shear stresses of the problem are written as:(16)$$\tau_{rz}=\mu_{mf}{\left(\frac{\partial w}{\partial r}\right)}_{r=R}\;and\;\;\tau_{r\theta }=\mu_{mf}{\left(\frac{\partial v}{\partial r}\right)}_{r=R}$$
which finally are written in the form: (17)τrz=μmf aRez1/2 f″0  and  τrθ=μmfaδRez1/2ζg′0
where Rez1/2=az2υf is the local Reynolds number, $\zeta =\frac{z}{R}$ is the dimensionless axial coordinate, and $\delta =\frac{a}{\omega }$ is the ratio between the stretching rate over the rate of an angular velocity of the cylinder. 

**Rate of heat transfer:** The heat transfer rate of the surface is obtained as: (18)Nu=zκmfκf(Tc−Tw)∂T∂rr=R=−κmfκfRez1/2θ′0

## 5. Computational Technique

This section represents the well-known Kafoussias et al. [38] two-point boundary layer point numerical approach. This numerical technique has a few key components that make it simple, accurate, and efficient. Potential characteristics include:(i)It is based on common finite difference with central differencing.(ii)Manipulation of tridiagonal matrices(iii)Implementation of an iterative procedure.

Equation (12) can be considered as a second-order linear differential equation by setting $F\left(x\right)=f^{\prime }\left(\eta \right),\;F^{\prime }\left(x\right)=f^{\prime \prime }\left(\eta \right),\;F^{\prime \prime }\left(x\right)=f^{\prime \prime \prime }\left(\eta \right)$, provided that $f\left(\eta \right)$ is considered a known function. In this case, momentum Equation (12) can be written as:$$\begin{array}{l} \left(1+2\eta D\right)\;\left(1+\frac{3}{2}\varphi \;m\right)\;F^{\prime \prime }+\left[2D\;\left(1+\frac{3}{2}\varphi \;m\right)+{\left(1-\varphi \right)}^{2.5}\;\left(1-\varphi +\varphi \;\frac{\rho_{s}}{\rho_{f}}\right)\;f\right]\;F^{\prime }\\ \;\;\;\;\;\;\;\;\;\;\;\;\;\;\;\;\;\;\;\;\;\;\;\;\;\;\;\;\;\;\;\;\;\;\;\;\;\;\;\;\;-{\left(1-\varphi \right)}^{2.5}\;\left(1-\varphi +\varphi \;\frac{\rho_{s}}{\rho_{f}}\right)\;f\;F+{\left(1-\varphi \right)}^{2.5}\beta \theta =0 \end{array}$$

Which is of the following form:(19)$$P\;\left(x\right)\;F^{\prime \prime }\left(x\right)+Q\left(x\right)F^{\prime }\left(x\right)+R\left(x\right)\;F\left(x\right)=S\left(x\right)\;$$
where
$$\begin{array}{l} P\left(x\right)=\left(1+2\eta D\right)\;\left(1+\frac{3}{2}\varphi \;m\right)\;,\;\;Q\left(x\right)=\left[2D\;\left(1+\frac{3}{2}\varphi \;m\right)+{\left(1-\varphi \right)}^{2.5}\;\left(1-\varphi +\varphi \;\frac{\rho_{s}}{\rho_{f}}\right)\;f\right]\;\\ R\left(x\right)\;=-\;{\left(1-\varphi \right)}^{2.5}\;\left(1-\varphi +\varphi \;\frac{\rho_{s}}{\rho_{f}}\right)\;\;f,\;\;S\left(x\right)=-\;{\left(1-\varphi \right)}^{2.5}\beta \theta \end{array}$$

The process start with guessing the initial conditions of $f^{\prime }\left(\eta \right)\;,\;g\left(\eta \right)\;,\;\theta \left(\eta \right)$ between η=0 and $\eta =\eta_{\infty }$ so that the boundary conditions (15) can be automatically satisfied. For that, we consider the following initial guesses:$$f^{\prime }\left(\eta \right)\;=\;g\left(\eta \right)\;=\;\theta \left(\eta \right)=1-\frac{\eta }{\eta_{\infty }}.$$

The curve $f\left(\eta \right)$ is determined by integrating of $f^{\prime }\left(\eta \right)$, where we assumed that f, g, θ are known and calculate new estimations of $f^{\prime }\left(\eta \right),\;f_{new}^{\prime }\left(\eta \right)$. Such values are used as fresh inputs, and, therefore, the corresponding profiles are updated and so on. This process is continued until the required convergence is attained using iterations since Equation (19) is non linear. 

The same algorithm is applied for Equations (13) and (14) as straightforward as when we obtained the profile of $f^{\prime }\left(\eta \right)\;$ but without iterations, since Equations (13) and (14) are linear. 

By setting $F\left(\eta \right)=g\left(\eta \right)\;$, Equation (13) takes the following form:$$\begin{array}{l} {\left(1+2\eta D\right)}^{2}\;\left(1+\frac{3}{2}\varphi \;m\right)\;F^{\prime \prime }+\left[2D\;\left(1+\frac{3}{2}\varphi \;m\right)\;\left(1+2\eta D\right)+{\left(1-\varphi \right)}^{2.5}\;\left(1-\varphi +\varphi \;\frac{\rho_{s}}{\rho_{f}}\right)\;\left(1+2\eta D\right)\;f\right]\;F^{\prime }\\ \;\;\;\;\;\;\;\;\;\;\;\;\;\;\;\;\;\;\;\;\;\;\;\;+\left[{\left(1-\varphi \right)}^{2.5}\;\left(1-\varphi +\varphi \;\frac{\rho_{s}}{\rho_{f}}\right)\;Df-\left(1+\frac{3}{2}\varphi \;m\right)\;D^{2}\right]F=0 \end{array}$$
which is already of the second order form and may be written as the previous form:(20)$$P\;\left(x\right)\;F^{\prime \prime }\left(x\right)+Q\left(x\right)F^{\prime }\left(x\right)+R\left(x\right)\;F\left(x\right)=S\left(x\right)\;$$
where
$$\begin{array}{l} P\left(x\right)={\left(1+2\eta D\right)}^{2}\;\left(1+\frac{3}{2}\varphi \;m\right)\;,Q\left(x\right)=\left[2D\;\left(1+\frac{3}{2}\varphi \;m\right)\;\left(1+2\eta D\right)+{\left(1-\varphi \right)}^{2.5}\;\left(1-\varphi +\varphi \;\frac{\rho_{s}}{\rho_{f}}\right)\;\left(1+2\eta D\right)\;f\right]\;\\ R\left(x\right)=\left[{\left(1-\varphi \right)}^{2.5}\;\left(1-\varphi +\varphi \;\frac{\rho_{s}}{\rho_{f}}\right)\;Df-\left(1+\frac{3}{2}\varphi \;m\right)\;D^{2}\right],S\left(x\right)=0 \end{array}$$

Similarly, by setting this time $F\left(\eta \right)=\theta \left(\eta \right)$, the temperature Equation (14) also may be written as: $$\begin{array}{l} \left(1+2\eta D\right)\;F^{\prime \prime }+\left[2D+\frac{\kappa_{f}}{\kappa_{mf}}\left(1-\varphi +\varphi \frac{{\left(\rho C_{p}\right)}_{s}}{{\left(\rho C_{p}\right)}_{f}}\right)\mathrm{Pr}f\right]F^{\prime }-\frac{\kappa_{f}}{\kappa_{mf}}\beta \;Ec_{2}f^{\prime }F\\ +\frac{\kappa_{f}}{\kappa_{mf}}\left[\beta \;\varepsilon \;\lambda \;f^{\prime }-\frac{\left(1+2\eta D\right)}{{\left(1-\varphi \right)}^{2.5}}Ec_{1}{\left\{g^{\prime }-\frac{D}{\left(1+2\eta D\right)}g\right\}}^{2}-\frac{\left(1+2\eta D\right)}{{\left(1-\varphi \right)}^{2.5}}\;Ec_{2}{f^{\prime \prime }}^{2}\right]=0 \end{array}$$
which is also of the form: (21)$$P\;\left(x\right)\;F^{\prime \prime }\left(x\right)+Q\left(x\right)F^{\prime }\left(x\right)+R\left(x\right)\;F\left(x\right)=S\left(x\right)\;$$
where
$$\begin{array}{l} P\left(x\right)=\left(1+2\eta D\right)\;,\;Q\left(x\right)=\left[2D+\frac{\kappa_{f}}{\kappa_{mf}}\left(1-\varphi +\varphi \frac{{\left(\rho C_{p}\right)}_{s}}{{\left(\rho C_{p}\right)}_{f}}\right)\mathrm{Pr}f\right],\;R\left(x\right)=-\frac{\kappa_{f}}{\kappa_{mf}}\beta \;Ec_{2}f^{\prime }\\ S\left(x\right)=-\frac{\kappa_{f}}{\kappa_{mf}}\left[\beta \;\varepsilon \;\lambda \;f^{\prime }-\frac{\left(1+2\eta D\right)}{{\left(1-\varphi \right)}^{2.5}}Ec_{1}{\left\{g^{\prime }-\frac{D}{\left(1+2\eta D\right)}g\right\}}^{2}-\frac{\left(1+2\eta D\right)}{{\left(1-\varphi \right)}^{2.5}}\;Ec_{2}{f^{\prime \prime }}^{2}\right] \end{array}$$

Thus, a new profile of $g\left(\eta \right)\;,g_{new}\left(\eta \right)\;$ is known as considered f is known. Similarly, considering that $f,f^{\prime },\;g,\;g^{\prime }\;$ are known functions, a new profile of $\theta \left(\eta \right)\;,\theta_{new}\left(\eta \right)\;$ is obtained. These two process continued until the required convergence is attained. 

For the numerical solution, we used the step size Δη=0.01, and by the error and trial solution, we fixed the value of $\eta_{\max}=10$, and we set the convergence criterion $10^{-5}$.

It should be noted that practically all of the PC’s RAM memory was utilized in order to achieve accuracy numbers that were close to five digits. We were therefore unable to solve the system of equations under consideration with much more accuracy than five digits due to memory capacity issues. In this scenario, using virtual memory would be pointless due to the significant increase in processing time. As a result, the memory-minimizing common finite difference method will be helpful, particularly for multi-dimensional issues that arise in fluid mechanics. The Intel (R) Core (TM) i3-6100 CPU with 8GB of RAM and a 64-bit operating system is used to calculate the numerical solution to the situation at hand.

## 6. Numerical Validation and Parameters Value Estimation

To establish the accuracy of our numerical scheme, we have compared our obtained results with the existing work of Munawar et al. [32], and these comparisons are depicted in Table 2. This gives us reliance of our computational outcomes.

Since we have studied the behavior of blood flow with CoFe_2_O_4_ magnetic particles, it is essential to put some realistic values in numerical process. For that, we have studied the relevant model and utilized the following values as captured in Table 3 and Table 4. Additionally, we considered the human body temperature $T_{w}$ = 37 °C, while body Curie temperature is $T_{c}=41$ °C, following studies [39,40]. Utilizing these values, the dimensionless temperature parameter is $\varepsilon =\frac{T_{c}}{T_{c}-T_{w}}=\frac{314}{314-310}=78.5$ [37]. The value of the Prandtl number for human body blood is $\mathrm{Pr}=\frac{{\left(\mu C_{p}\right)}_{f}}{\kappa_{f}}=\frac{3.2\times 10^{-3}\times 3.9\times 10^{3}}{0.5}≅25$, as the value of blood density is ρ=1050  Kgm^−3^, viscosity $\mu =3.2\times 10^{-3}$ Kgm^−1^s^−1^, thermal conductivity κ=0.5  Wm^−1^K^−1^, specific heat $C_{p}=3.9\times 10^{3}\;$ JKg^−1^K^−1^ stated in the literature [32,34,35], and viscous dissipation parameter $\lambda =6.4\times 10^{-14}$. Table 3 shows the values of blood and CoFe_2_O_4_ magnetic particles, whereas Table 4 shows those values used in numerical computation. 

## 7. Results and Discussion

After the application of the aforementioned algorithm, the obtained results are analyzed to explore the behavior of blood-CoFe_2_O_4_ particles up to a 10% volume fraction and pure blood flow model. The corresponding numerical results are presented through figures for the various values of control dimensionless parameters such as the ferromagnetic parameter, curvature parameter, effective magnetic number, Prandtl number, and Eckert number due to rotation and stretching of the cylinder.

The influence of the ferromagnetic number on axial velocity $f^{\prime }\left(\eta \right)$, tangential velocity $g\left(\eta \right)$, and temperature $\theta \left(\eta \right)$ distributions is shown in Figure 2, Figure 3 and Figure 4. It is observed that when particles of volume fraction are mixed with blood, the velocity of blood-CoFe_2_O_4_ is slightly improved compared to that of pure blood $\left(\varphi =0\right)$, except β=0. Here, due to magnetization forces in the flow domain, one kind of resistive force is induced, known as the Kelvin force, which causes a reduction in flow in the boundary layer. But due to rotational viscosity, we see from Figure 2, Figure 3 and Figure 4 that both axial velocity and temperature are reduced, while tangential velocity increases in this case. 

Figure 5, Figure 6 and Figure 7 illustrate the impacts of curvature parameter D on the $f^{\prime }\left(\eta \right)$, $g\left(\eta \right)$ and $\theta \left(\eta \right)$ profiles, respectively. It is observed that with enhancing values of D, the axial velocity and temperature distributions are accelerated but decline in the tangential velocity profile. It is also noted that blood-CoFe_2_O_4_ has better velocity distributions compared to that of pure blood, whereas the reverse behavior is observed in the temperature distribution. The fact is that as D increases, ultimately, the radius of the cylinder decreases, which is caused by the improvement in the thermal boundary layer as well as temperature profile enhancement. 

Figure 8, Figure 9 and Figure 10 show the effect of the effective magnetic number on axial velocity, tangential velocity, and temperature distributions, respectively. When a magnetic field is applied, magnetic particles and blood rotate at various angular velocities in a biofluid rotational flow. The rotating flow of biofluid experiences an increase in viscosity as a result of this discrepancy. The tangential and axial velocities increase in Figure 8 and Figure 9 as the effective magnetic parameter increases. The influence of magnetic field intensity on the fluid grows and becomes more dominant as the magnetic particles in biofluid get closer to each other. The magnetic fluid rotates more favorably due to the magnetic field. The temperature distribution for raising the values of the effective magnetization parameter is shown and is not significant as a velocity profile.

The profiles of the temperature are illustrated in Figure 11, Figure 12 and Figure 13 to determine the effects of the Eckert number $Ec_{1}$, $Ec_{2}$ and Prandtl number, respectively. It is evident from Figure 11 and Figure 12 that both temperature profiles decrease with accelerating values of the Eckert number. However, degeneration in the temperature distribution for a rotating cylinder $Ec_{1}$ is more cabbalistic as compared to that of a stretching Eckert number $Ec_{2}$ due to the presence of a magnetic field and the rotational viscosity behavior of the fluid. Consequently, the impacts of various values of the Prandtl number on the velocity profile are shown in Figure 13, and it was found that the temperature diminishes as the Prandtl number increases, which indicates that the viscosity of the thermal boundary layer falls in opposition to the Prandtl number, since larger values of the Prandtl number relate to sickly thermal diffusivity and cause a sleazy thermal boundary layer. 

Finally, the skin friction coefficient in axial and tangential directions and the rate of heat transfer are presented in Figure 14, Figure 15, Figure 16, Figure 17, Figure 18 and Figure 19 for various values of the ferromagnetic parameter and the curvature parameter. It is found that blood-CoFe_2_O_4_ gives a slightly better $-\theta^{\prime }\left(0\right)$ and $g^{\prime }\left(0\right)$ performance than pure blood, whereas a reverse trend is observed in $f^{\prime \prime }\left(0\right)$. As both β and D raise, the coefficient of skin friction in the axial direction and the rate of heat transfer are enhanced while $g^{\prime }\left(0\right)$ decreases. This is due to the combined effect of the Kelvin force and the radius of the cylinder in the flow domain. Because the Kelvin force is stronger in the blood flow domain than the Lorentz force, blood flows through surfaces with a cylindrical shape at a faster rate of heat transfer, as seen in Figure 16.

## 8. Conclusions

The steady, viscous flow of biomagnetic fluid containing magnetic particles over a rotating stretchable cylinder is analyzed in this study. A well-established finite difference algorithm has been utilized to solve the developed mathematical physical model for temperature and velocity distributions along with physical quantities under the influence of well-known parameters. From the present investigation, we can draw the following conclusions: (i)With rising values of the ferromagnetic interaction parameter and effective magnetic number, the tangential velocity is reduced, whereas the axial and temperature distributions are accelerated.(ii)We observe a favorable behavior in cases of adding particles to the base fluid compared to that of pure fluid.(iii)As the values of the curvature parameter rise, both axial velocity and temperature increase.(iv)The heat transfer of fluid decline as the values of the Prandtl number and Eckert numbers rise.(v)For larger values of the ferromagnetic interaction parameter and the curvature parameter, the skin friction coefficient and the rate of heat transfer increase.

## Figures and Tables

**Figure 1 bioengineering-11-00317-f001:**
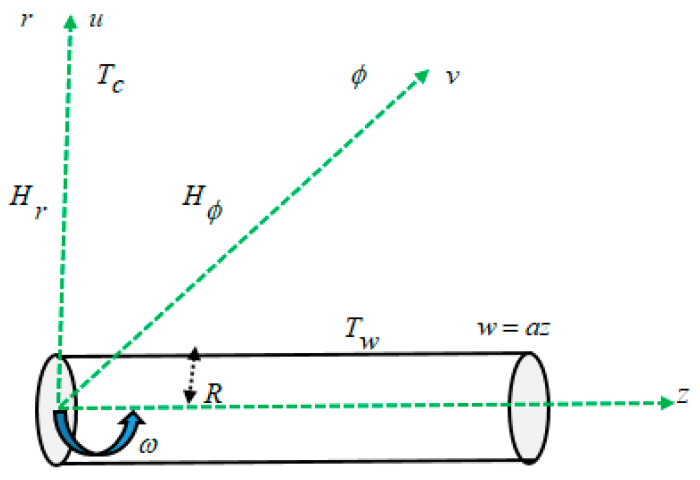
Geometry view of the flow of present study.

**Figure 2 bioengineering-11-00317-f002:**
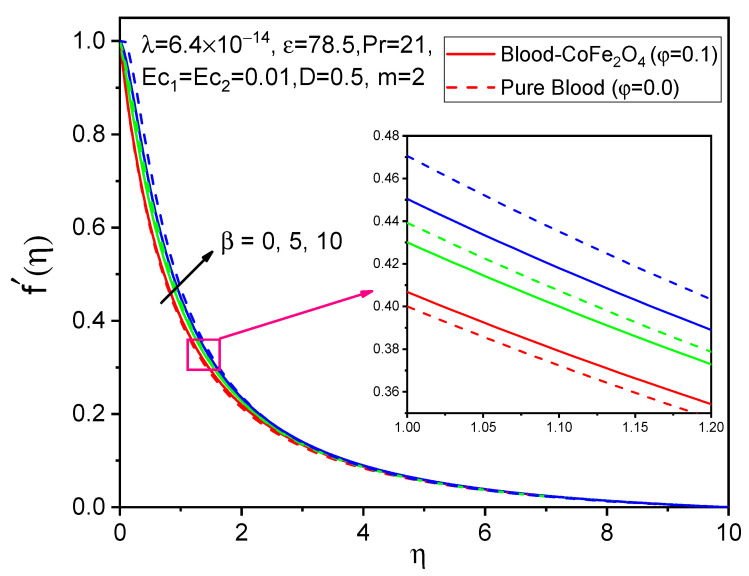
Effect of β on $f^{\prime }\left(\eta \right)$.

**Figure 3 bioengineering-11-00317-f003:**
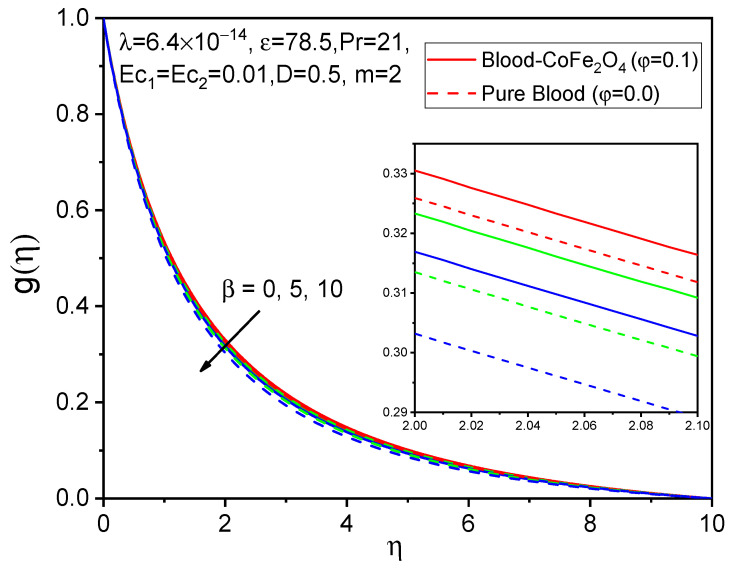
Effect of β on $g\left(\eta \right)$.

**Figure 4 bioengineering-11-00317-f004:**
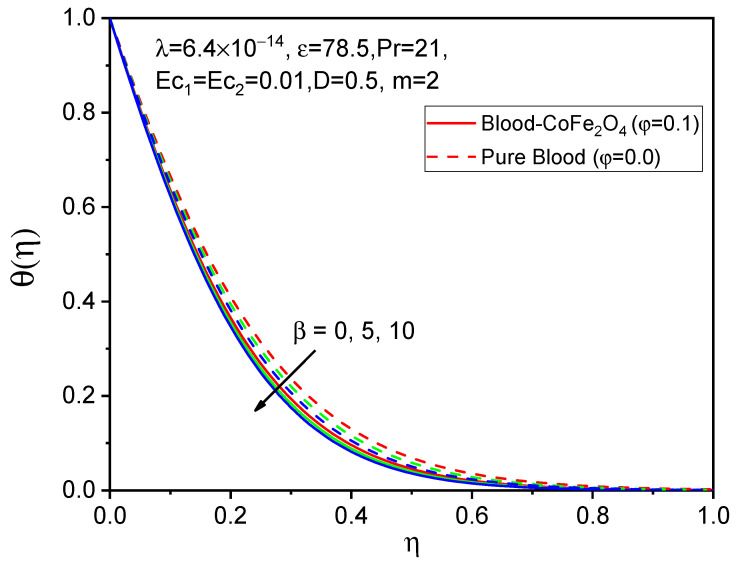
Effect of β on $\theta \left(\eta \right)$.

**Figure 5 bioengineering-11-00317-f005:**
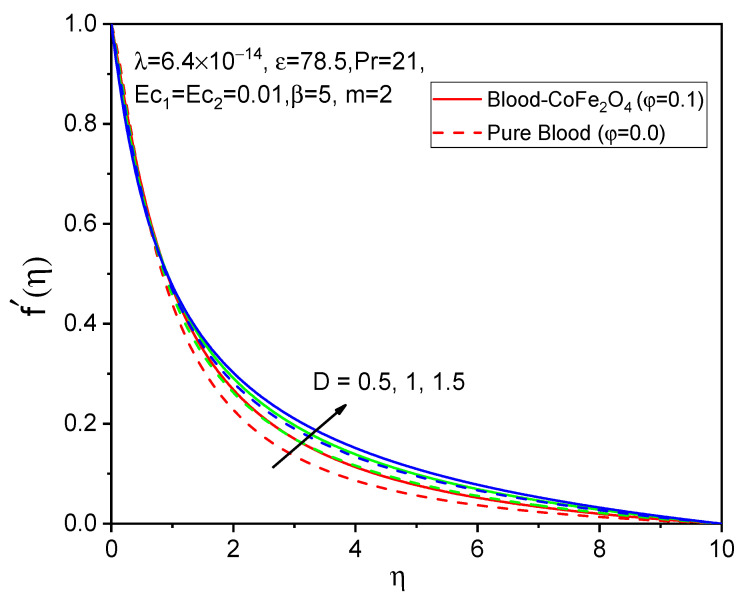
Effect of D on $f^{\prime }\left(\eta \right)$.

**Figure 6 bioengineering-11-00317-f006:**
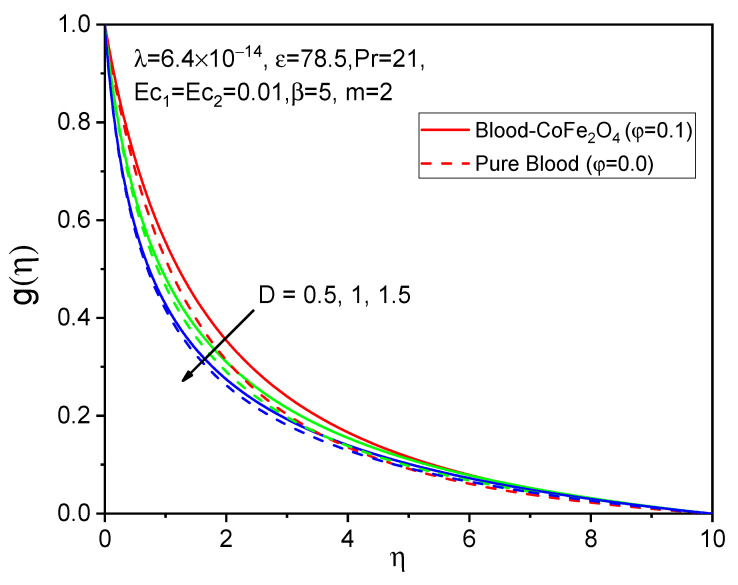
Effect of D on $g\left(\eta \right)$.

**Figure 7 bioengineering-11-00317-f007:**
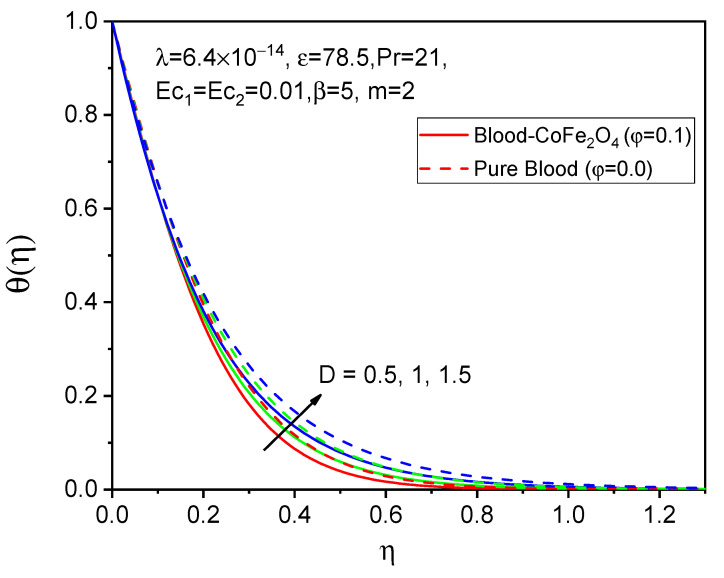
Effect of D on $\theta \left(\eta \right)$.

**Figure 8 bioengineering-11-00317-f008:**
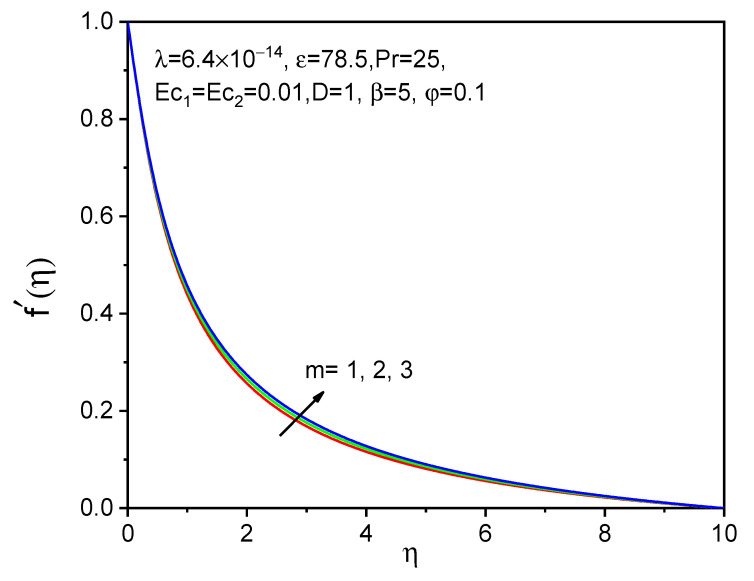
Effect of m on $f^{\prime }\left(\eta \right)$.

**Figure 9 bioengineering-11-00317-f009:**
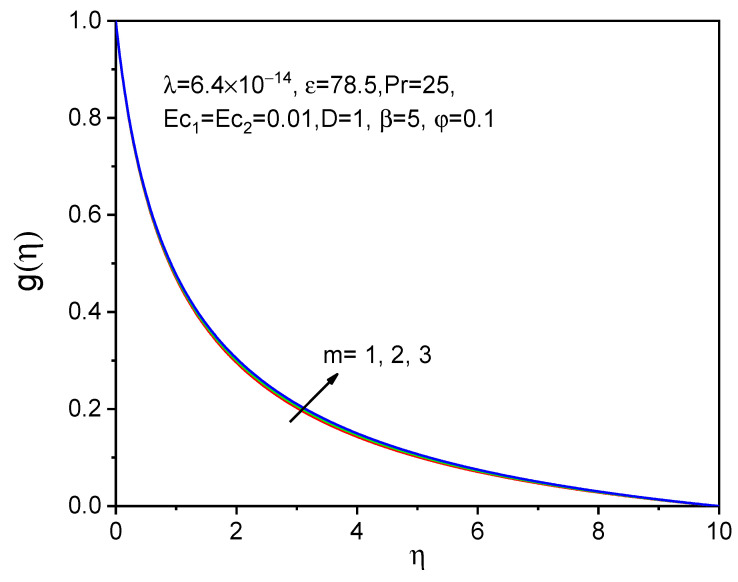
Effect of m on $g\left(\eta \right)$.

**Figure 10 bioengineering-11-00317-f010:**
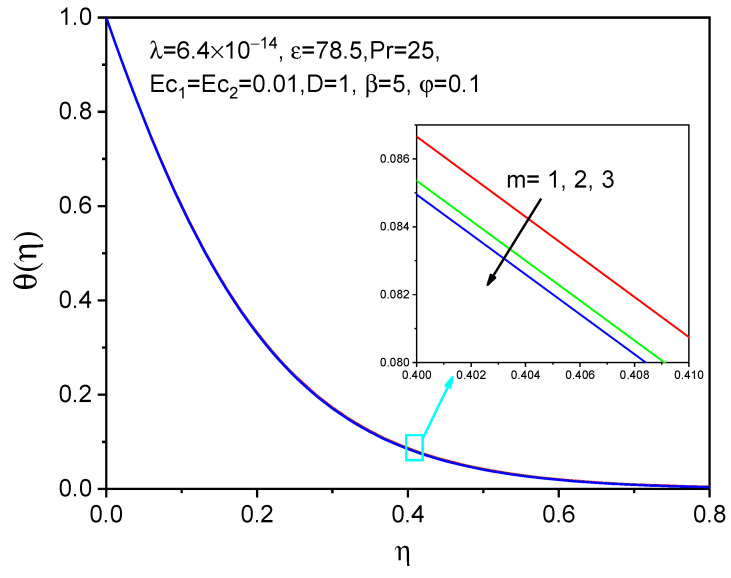
Effect of m on $\theta \left(\eta \right)$.

**Figure 11 bioengineering-11-00317-f011:**
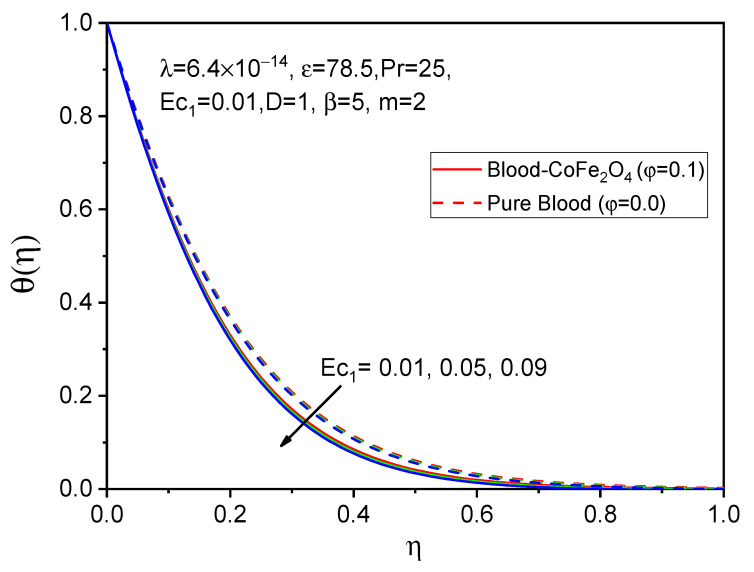
Effect of $Ec_{1}$ on $\theta \left(\eta \right)$.

**Figure 12 bioengineering-11-00317-f012:**
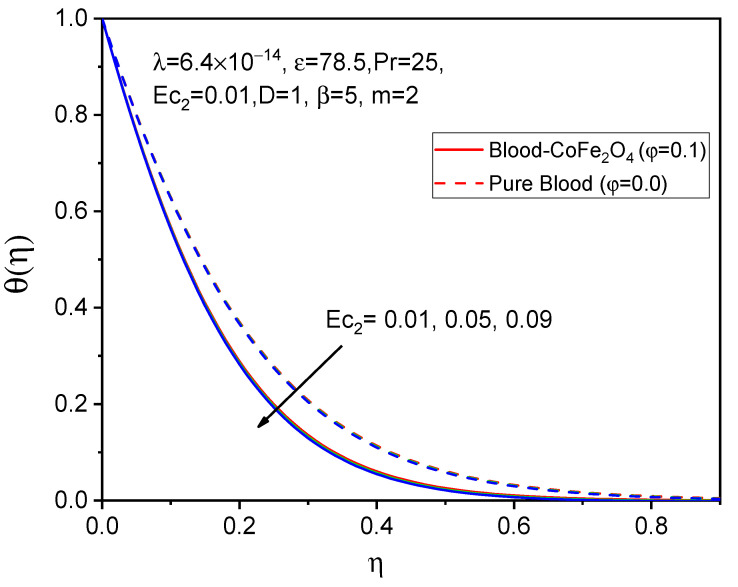
Effect of $Ec_{2}$ on $\theta \left(\eta \right)$.

**Figure 13 bioengineering-11-00317-f013:**
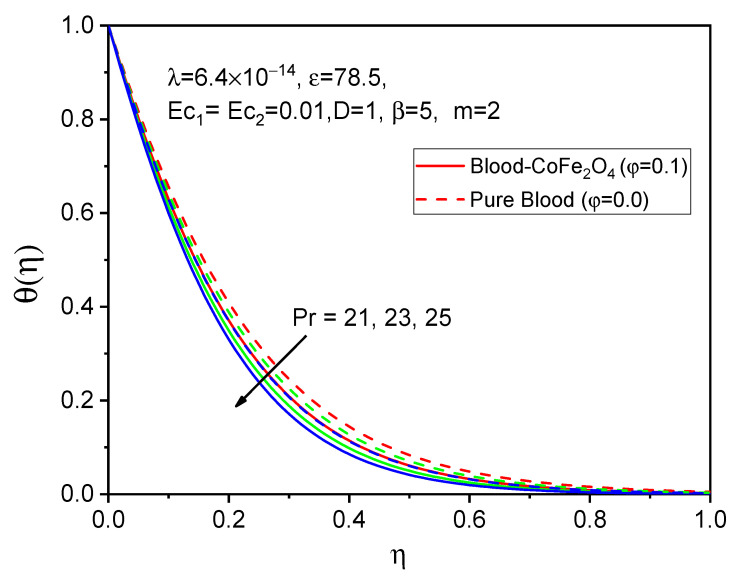
Effect of Pr on $\theta \left(\eta \right)$.

**Figure 14 bioengineering-11-00317-f014:**
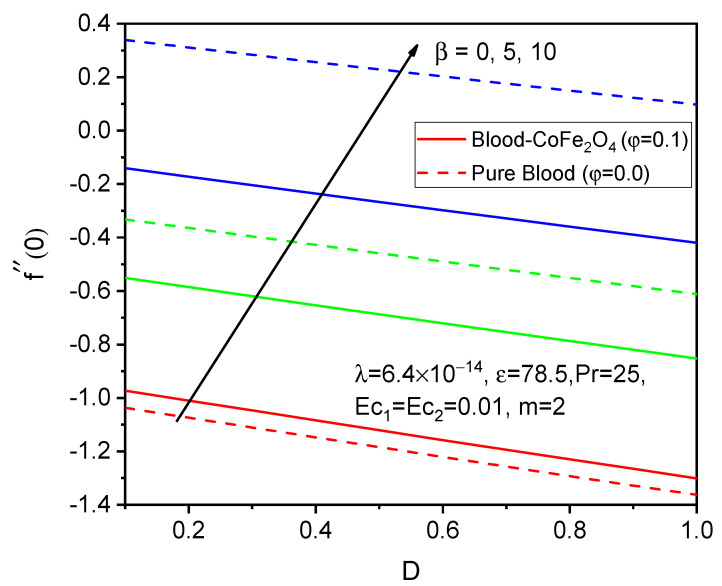
Variations in $f^{\prime \prime }\left(0\right)$ for various values of β against D.

**Figure 15 bioengineering-11-00317-f015:**
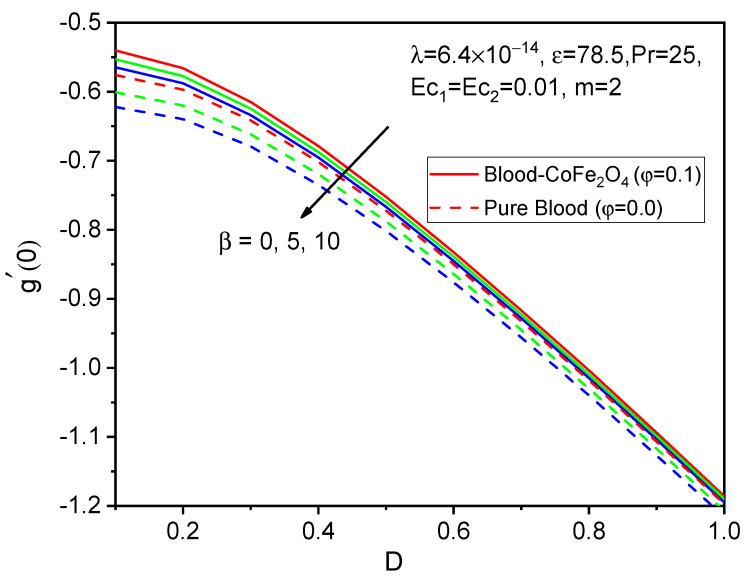
Variations in $g^{\prime }\left(0\right)$ for various values of β against D.

**Figure 16 bioengineering-11-00317-f016:**
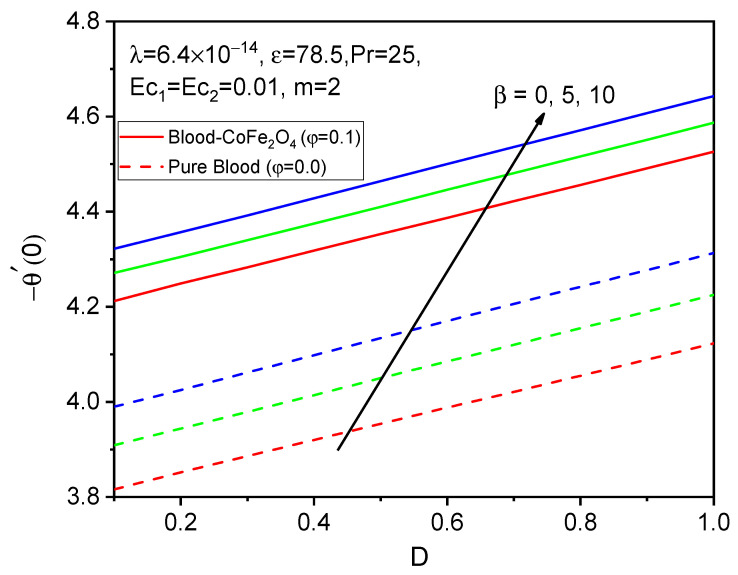
Variations in $-\theta^{\prime }\left(0\right)$ for various values of β against D.

**Figure 17 bioengineering-11-00317-f017:**
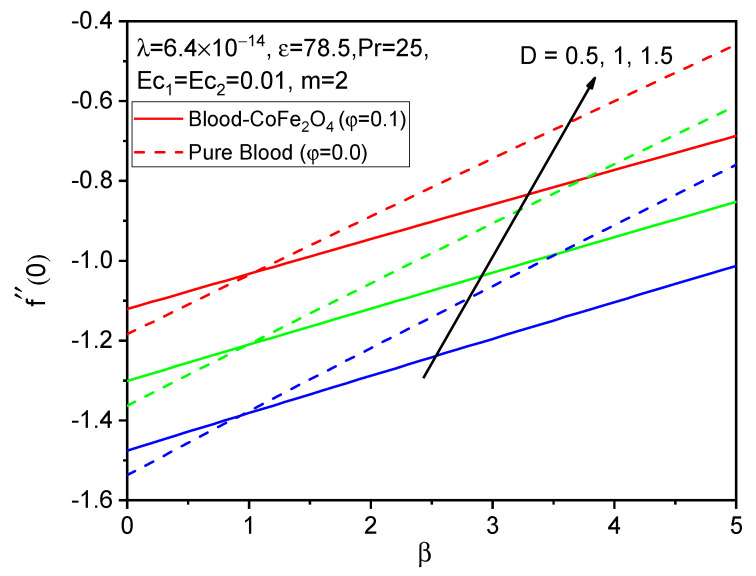
Variations in $f^{\prime \prime }\left(0\right)$ for various values of D against β.

**Figure 18 bioengineering-11-00317-f018:**
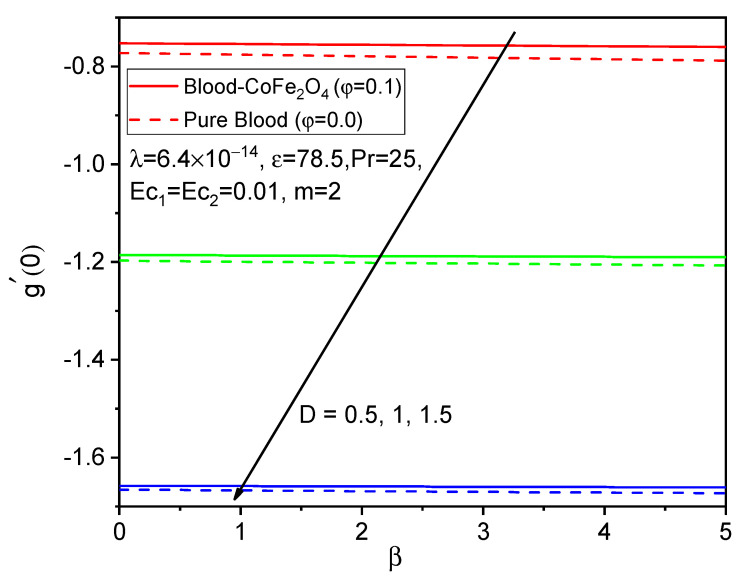
Variations in $g^{\prime }\left(0\right)$ for various values of D against β.

**Figure 19 bioengineering-11-00317-f019:**
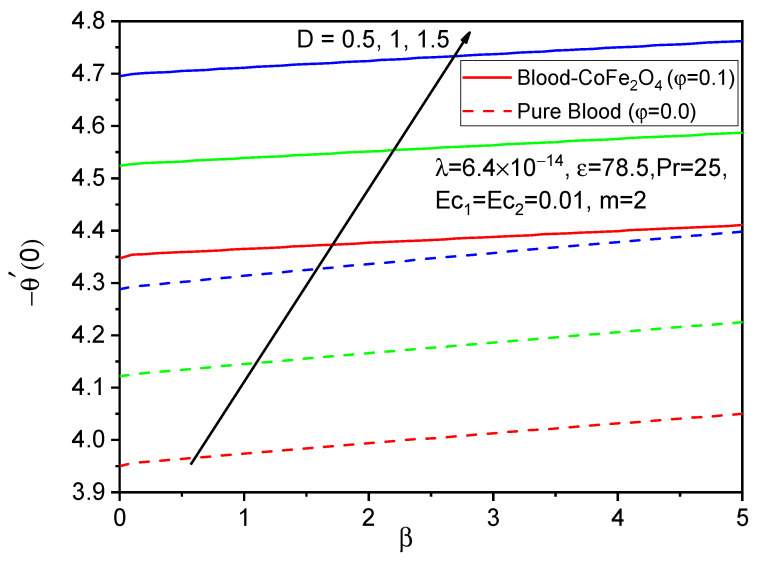
Variations in $-\theta^{\prime }\left(0\right)$ for various values of D against β.

**Table 1 bioengineering-11-00317-t001:** Theoretical correlation for the magnetic fluid [36,37].

Density	$\rho_{mf}=\left(1-\varphi \right)\rho_{f}+\varphi \rho_{s}$
Dynamic viscosity	$\mu_{mf}=\mu_{f}{\left(1-\varphi \right)}^{-2.5}$
Heat capacity	${\left(\rho C_{p}\right)}_{mf}=\left(1-\varphi \right)\;{\left(\rho C_{p}\right)}_{f}+\varphi {\left(\rho C_{p}\right)}_{s}$
Thermal conductivity	$\frac{\kappa_{mf}}{\kappa_{f}}=\frac{\left(\kappa_{s}+2\kappa_{f}\right)-2\varphi \left(\kappa_{s}+2\kappa_{f}\right)}{\left(\kappa_{s}+2\kappa_{f}\right)+\varphi \left(\kappa_{s}+2\kappa_{f}\right)}$

**Table 2 bioengineering-11-00317-t002:** Comparison values of rate of heat transfer $-\theta^{\prime }\left(0\right)$ for ordinary fluid with Munawar et al. [32] when $m=\varphi =\beta =0,\mathrm{Pr}=2,Ec_{1}=Ec_{2}=0.2$ for various values of curvature parameter.

D	Present Results	Munawar et al. [32]
0	0.65322	0.65421
0.5	0.58741	0.58756
2	0.25880	0.2578

**Table 3 bioengineering-11-00317-t003:** Thermo-physical values of blood and CoFe_2_O_4_ [20,21,41,42].

Properties	Base Fluid	Magnetic Particles
	Blood	CoFe_2_O_4_
$C_{p\;}$(JKg^−1^K^−1^)	$3.9\times 10^{3}$	700
ρ (Kgm^−3^)	1050	4907
κ (Wm^−1^K^−1^)	0.5	3.7

**Table 4 bioengineering-11-00317-t004:** Fixed values in computational process.

Parameter	Values	References
$\mathrm{Ferromagnetic}\;\mathrm{interaction}\;\mathrm{parameter}\;\left(\beta \right)$	0, 5, 10	[37,39]
$\mathrm{Curvature}\;\mathrm{parameter}\;\left(D\right)$	0.5, 1, 1.5	[20,21]
$\mathrm{Particles}\;\mathrm{volume}\;\mathrm{fraction}\;\left(\varphi \right)$	0, 0.1	[20,37,42]
$\mathrm{Effective}\;\mathrm{magnetic}\;\mathrm{number}\;\left(m\right)$	1, 2, 3	[34,36]
$\mathrm{Prandtl}\;\mathrm{number}\;\left(\mathrm{Pr}\right)$	21, 23, 25	[20,21,39]
$\mathrm{Eckert}\;\mathrm{number}\;\mathrm{due}\;\mathrm{to}\;\mathrm{rotational}\;\mathrm{cylinder}\;\left(Ec_{1}\right)$	0.01, 0.05, 0.09	[32,43]
$\mathrm{Eckert}\;\mathrm{number}\;\mathrm{due}\;\mathrm{to}\;\mathrm{rotational}\;\mathrm{cylinder}\;\left(Ec_{2}\right)$	0.01, 0.05, 0.09	[32,43]

## Data Availability

The data that support the findings of this study are available from the corresponding author.

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
