# Peer review of "Flow and Heat Transfer of CoFe2O4-Blood Due to a Rotating Stretchable Cylinder under the Influence of a Magnetic Field"

_bioengineering, 2024, doi:10.3390/bioengineering11040317_

Round 1

Reviewer 1 Report

Comments and Suggestions for Authors

The quality level of the presentation is definitely below the standard required for a scientific publication. This concern applies to both the literal and the mathematical presentation.

The paper needs to be consistently revised before it could be considered for publication, as noted down in the attached copy of the manuscript.

Comments on the Quality of English Language

The level of the literal presentation is below standard. I would suggest a revision guided by a professional editor/natural language speaker.

Author Response

Thank you for your comments concerning our manuscript. Those comments are all valuable and very helpful for revising and improving our paper, as well as the important guiding significance to our researches. We have studied comments carefully and made correction point by point.  

Reviewer’s comments and Response to comments:

  1. The quality level of the presentation is definitely below the standard required for a scientific publication. This concern applies to both the literal and the mathematical presentation. The paper needs to be consistently revised before it could be considered for publication, as noted down in the attached copy of the manuscript. Peer-review-33085832.v1.pdf .The level of the literal presentation is below standard. I would suggest a revision guided by a professional editor/natural language speaker

Answer: Thank you for your precious comments. Based on your suggestion, we revised the whole manuscript again and revised according to your concerns. All possible corrections are marked with yellow color in the revised manuscript in order to improve the standard of the manuscript for publications in bioengineering. We hope that the revised version will meet the standard of the journal. Please find the attached revised manuscript. 

We tried our best to improve the manuscript and made some changes to it. We sincerely thank you for your enthusiastic work and hope that the correction will be approved.

Reviewer 2 Report

Comments and Suggestions for Authors

The authors should address the following important points during the preparation of the revised paper

1                    Remove all typos from the text.

      2          The manuscript needs Nomenclature part

      3.     The introduction must be built in such a way that each paragraph addresses a major keyword of the problem. Enrich this part with more recent subject of hybrid nano‑blood through a ciliated micro‑vessel subject to .. lorenz force and other us Peristaltic activity in blood flow of Casson nanoliquid..

    4.       The equations of the computational technique must be well verified.

     5.     The discusion part is poor, this part must be well improved with physical interpretation

     6. What is the novelty of this research work?  Discuss this in the last paragraph of the introduction section.

    7. The old references must replaced with new ones (2021-2023)

Comments on the Quality of English Language

             This work is of interest subject to minor corrections of the minor English errors.

Reviewer 3 Report

Comments and Suggestions for Authors

The topic of the paper ``Flow and heat transfer of CoFe2O4-blood due to a rotating stretchable cylinder under the influence of a magnetic field'' by Jahangir Alam, Ghulam Murtaza, Efstratios Tzirtzilakis, Shuyu Sun and Mohammad Ferdows is of interest for scientists (especially numerical analysts) and doctors working with blood flows. However, to be published in a high bar Journal like the Bioengineering (Section: Biomechanics and Sports Medicine, Special Issue: Non-Newtonian Blood Flow in Computational Fluid Dynamics) where it is submitted, it should be radically reworked. I suggest the following improvements. 

line 208: the reference [28] is not sufficient, a brief description of the numerical methods used is needed.

Some of the sentences do not make sense, e.g., lines 214-215: ``Since, the equation (12) are highly nonlinear, so at first we reduce equation (12) in to 214 a 2nd order differential equation by letting,...'' such sentences should be carefully reformulated throughout the paper.

The question of generation of magnetic field by such electrically conducting fluid flows is not mentioned at all (in other words, why the magnetic induction equations is not a part of the model?).

It not clear for me in what the model and the results differ from the existing ones.  The authors should clarify this. 

On most of the figures the lines are extremely close to each other, making difficult the comparison. Also, I would suggest considering representing in Fig.1--13 the vertical axis in the logarithmic format.

The absence of information pertaining to the computer utilized for the numerical experiments in the paper introduces a notable gap in understanding. It leaves the readers uncertain about crucial details such as whether the computations were executed on a laptop, workstation, a cluster node, or perhaps another computing platform. This ambiguity can significantly impact the reproducibility and interpretation of the study's results. To address this issue, it is strongly recommended to incorporate comprehensive details about the computing environment used to produce results described in the paper. Providing information such as the specifications of the hardware and any relevant software configurations will not only enhance the transparency of the research but also serve as valuable insights for researchers aiming to replicate or build upon the presented findings. By offering a clear description of the computational infrastructure employed, the paper can contribute to the credibility and robustness of the reported numerical experiments, fostering a more thorough understanding within the scientific community.

To elevate the overall caliber of the manuscript, it would be beneficial for the author to offer a succinct discussion outlining their prospective research direction in this area. Notably, since it appears that all computations presented in the paper were likely carried out on a laptop, delving into the potential avenues for extending and customizing the elucidated techniques to confront more computationally demanding challenges would be enlightening. Providing insights into how the methodologies expounded upon can be broadened and adapted to tackle tasks of heightened computational complexity could significantly enrich the paper. One avenue worth exploring is the optimization of these methods for deployment on multiprocessor computers. This optimization has the potential to bolster their scalability and computational efficiency, thereby opening doors to more robust applications in scenarios demanding substantial computational resources. By addressing these considerations, the author can not only enhance the current paper's academic merit but also contribute valuable perspectives for the broader scientific community, encouraging further exploration and advancement in the field.

Comments on the Quality of English Language

Extensive editing of English language required.

Round 2

Reviewer 1 Report

Comments and Suggestions for Authors

The paper has been substantially improved and may certainly be published.

Comments on the Quality of English Language

I would suggest a fine polishing. For example, a sentence like "the velocity accelerates" does not sound completely correct from a logical standpoint.

Author Response

Thank you for your precious suggestion. Based on your suggestion, we revised the whole manuscript again and polish the manuscript with our best knowledge. We hope that the revised manuscript will be met the journal standard on the quality of English Language for publication. The polishing portions of the revised manuscript are marked with yellow colour.

We tried our best to improve the manuscript and made some changes to it. We sincerely thank you for your enthusiastic work and hope that the correction will be approved.

Thank you again for accepting the revised manuscript for publication in bioengineering journal.

Yours sincerely,

Mohammad Ferdows

Address: Research Group of Fluid Flow Modeling and Simulation, Department of Applied Mathematics, University of Dhaka, Dhaka-1000, Bangladesh

Reviewer 2 Report

Comments and Suggestions for Authors

Accept

Comments on the Quality of English Language

Accept

Author Response

Thank you for your comments and suggestions. Those comments/suggestions are all valuable and very helpful for revising and improving our paper, as well as the important guiding significance to our researches.

Thank you again for accepting the revised manuscript for publication in bioengineering journal.

Yours sincerely,

Mohammad Ferdows

Address: Research Group of Fluid Flow Modeling and Simulation, Department of Applied Mathematics, University of Dhaka, Dhaka-1000, Bangladesh

Reviewer 3 Report

Comments and Suggestions for Authors

The authors have answered all the point I had raised in my review. Now, in my opinion,  the paper can be accepted. 

Comments on the Quality of English Language

Minor editing of English language required.

Author Response

Thank you for your precious suggestion. Based on your suggestion, we revised the whole manuscript again and polish the manuscript with our best knowledge. We hope that the revised manuscript will be met the journal standard on the quality of English Language for publication. The polishing portions of the revised manuscript are marked with yellow colour.

Thank you again for accepting the revised manuscript for publication in bioengineering journal.

Yours sincerely,

Mohammad Ferdows

Address: Research Group of Fluid Flow Modeling and Simulation, Department of Applied Mathematics, University of Dhaka, Dhaka-1000, Bangladesh
